# Tortuous Cardiac Intercalated Discs Modulate Ephaptic Coupling

**DOI:** 10.3390/cells11213477

**Published:** 2022-11-02

**Authors:** Ena Ivanovic, Jan P. Kucera

**Affiliations:** Department of Physiology, University of Bern, CH-3012 Bern, Switzerland

**Keywords:** cardiac electrophysiology, intercalated discs, ephaptic coupling, sodium channels, computer modeling

## Abstract

Cardiac ephaptic coupling, a mechanism mediated by negative electric potentials occurring in the narrow intercellular clefts of intercalated discs, can influence action potential propagation by modulating the sodium current. Intercalated discs are highly tortuous due to the mingling of plicate and interplicate regions. To investigate the effect of their convoluted structure on ephaptic coupling, we refined our previous model of an intercalated disc and tested predefined folded geometries, which we parametrized by orientation, amplitude and number of folds. Ephaptic interactions (assessed by the minimal cleft potential and amplitude of the sodium currents) were reinforced by concentric folds. With increasing amplitude and number of concentric folds, the cleft potential became more negative during the sodium current transient. This is explained by the larger resistance between the cleft and the bulk extracellular space. In contrast, radial folds attenuated ephaptic interactions and led to a less negative cleft potential due to a decreased net cleft resistance. In conclusion, despite limitations inherent to the simplified geometries and sodium channel distributions investigated as well as simplifications regarding ion concentration changes, these results indicate that the folding pattern of intercalated discs modulates ephaptic coupling.

## 1. Introduction

The synchronous contraction of the heart is coordinated by propagating electrical signals known as action potentials. Alterations in cardiac action potential propagation can lead to severe cardiac arrhythmias, including ventricular fibrillation, which can lead to sudden cardiac death. The action potential is known to propagate due to low-resistance pathways between neighboring cardiomyocytes formed by gap junctional channels [1,2,3,4]. However, it had already been suggested in the 1970s that a second mechanism may be involved in cardiac conduction [5]. This mechanism is called ephaptic coupling. Ephaptic coupling refers to intercellular communication mediated by extracellular electric fields and potentials through close membrane juxtaposition [6]. Hence, it relies on negative electrical potentials occurring in intercellular clefts of intercalated discs. These negative cleft potentials are caused by the large inward sodium (Na^+^) current (I_Na_) in the intercalated disc membranes, which necessarily also flows through the narrow extracellular space of the intercalated discs. These negative extracellular potentials exert feedback on the Na^+^ current via the mechanisms of self-attenuation, self-activation [7,8,9,10,11,12,13,14,15] and Na^+^ transfer [16]. The term self-attenuation describes the reduction of the Na^+^ current in the pre-junctional membrane due to reduced driving force. In contrast, the term self-activation describes faster activation of Na^+^ channels in the pre-junctional membrane and activation of Na^+^ channels in the post-junctional membrane (transactivation) due to a less negative transmembrane potential, i.e., a membrane potential that is brought closer, or even over, the Na^+^ channel activation threshold. Thus, self-activation can be considered as the gating (activation) of Na^+^ channels into their open state. Both mechanisms, self-activation and self-attenuation, may occur concomitantly and have a positive, but also a negative effect on action potential propagation. While narrow clefts (with high radial resistance) typically lead to self-activation and transactivation, self-attenuation then occurs because the extracellular cleft potential becomes so negative that post-junctional I_Na_ remains low despite full transactivation [14]. The concept of Na^+^ transfer, which we characterized in recent work [16], describes the switch of the pre-junctional Na^+^ current from an inward to an outward current, which possibly provides the cleft with additional Na^+^ ions to depolarize the post-junctional cell. Thus, the modulation of the Na^+^ current by ephaptic interactions can influence action potential propagation, particularly when gap junctional coupling is reduced [5,8,14,15,16,17,18].

For ephaptic coupling to occur, two conditions are of great importance: narrow intermembrane spacing between adjacent cells as well as a high density of Na^+^ channels in intercalated disc membranes [5,7,8,10,17]. The narrowness of the cleft is well established [12,19], and the high density of Na^+^ channels has been demonstrated by several studies [20,21,22,23]. Furthermore, in cardiac intercalated discs, Na^+^ channels form clusters in the vicinity of gap junction plaques [12,24,25], forming the basis of cardiac ephapses. Moreover, Na^+^ channels cluster around N-cadherin [23], and such clusters can also be considered as functional excitability nodes. In previous computational studies, we demonstrated that the high density of Na^+^ channels located in intercalated disc membranes [7] and the formation of Na^+^ channel clusters enhance ephaptic coupling [14] and thus facilitate action potential propagation for reduced gap junctional coupling. More recently, using a newly developed three-dimensional computational modeling framework of two longitudinally abutting cardiomyocytes, we showed that intercalated disc nanodomains around gap junction plaques, known as perinexi, are privileged sites for ephaptic coupling [16]. In both modeling studies [14,16], the intercalated disc was represented in high resolution, but the simplifying assumption was made that it was flat.

However, from microscopy studies, it is known that cardiac intercalated discs have a highly tortuous structure due to the mingling of plicate and interplicate regions [23,26,27,28,29,30,31,32,33]. In addition, plicate regions exhibit characteristic nanoscopic foldings [29,30,31,33]. Recently, Moise et al. [34] developed an algorithm to generate realistic high-resolution meshes (with approximately 300,000 nodes) of the intercalated disc based on transmission electron microscopy images, which included intermembrane spacing, gap junction plaques and plicate and interplicate regions. In their study, they subsequently incorporated the complex intercalated disc structure into a cardiac conduction model (a fiber of cells) and observed that intercalated disc morphology modulated conduction via ephaptic coupling. However, integration of the high-resolution meshes into the conduction model necessitated their reduction to networks of approximately 200 nodes coupled via electrical resistors. To date, the electrophysiological relevance of intercalated disc tortuosity is still not well understood.

Therefore, the aim of the present study was to investigate how different convoluted intercalated disc structures modulate ephaptic coupling. For this purpose, we refined our previously developed finite element model of an intercalated disc shared by two cardiomyocytes [14] and tested different predefined folded intercalated disc geometries, which we parametrized by the orientation of folds (concentric vs. radial) and by the amplitude and number of folds. In our simulations, Na^+^ channels were either uniformly distributed or clustered in the center of the intercalated discs.

We found that for both Na^+^ channel distributions, concentric folds reinforced ephaptic interactions with increasing amplitude and number of folds. This can be explained by a longer average path from a point in the cleft to the bulk extracellular space, which increases the net radial resistance of the cleft and leads to a more negative cleft potential during the Na^+^ current transient. In contrast, ephaptic interactions were attenuated with increasing amplitude and number of radial folds, although this effect was smaller. This attenuation was explained by more possible paths from a point in the cleft to the bulk extracellular space, which decreases net radial cleft resistance. In the presence of Na^+^ channel clusters, concentric and radial folds modulated ephaptic coupling in a more complex manner. In summary, the major impact of different intercalated disc geometries is mediated by a change in radial resistance towards the bulk interstitium. By using predefined folded intercalated disc geometries, our study thus represents a first step towards a better understanding of the electrophysiological role of tortuous intercalated discs in action potential propagation with ephaptic coupling.

## 2. Methods

### 2.1. Computational Model

Our computational model is based on the high-resolution finite element modeling framework of an intercalated disc shared by two cardiomyocytes, which was developed by Hichri et al. [14]. In this model, charge conservation is defined as in Equation (1):(1)$$-\nabla \left({w\;\sigma }_{e}\;\nabla V_{e}\right)=I_{\mathrm{ion},1}+C_{m}\frac{\partial V_{m,1}}{\partial t}+I_{\mathrm{ion},2}+C_{m}\frac{\partial V_{m,2}}{\partial t}$$
(2)$$V_{m,1}=V_{i,1}-V_{e}$$
(3)$$V_{m,2}=V_{i,2}-V_{e}$$
with w being cleft width, σ_e_ representing the extracellular conductivity tensor, C_m_ representing membrane capacitance (per unit area), and I_ion,1_ and I_ion,2_ representing the ion current densities through the two neighboring membranes. V_m,1_ and V_m,2_ represent the respective membrane potentials calculated as the potential difference between the intracellular (V_i,1_ and V_i,2_) and extracellular potential V_e_ (Equations (2) and (3)). In the present study, the two-dimensional flat finite element framework by Hichri et al. [14] was extended by a third spatial coordinate, such that the intercalated disc is now considered as a curved two-dimensional surface (manifold) embedded in three dimensions. Hence, the operator ∇ must be understood as the gradient operator in the tangent space of the curved intercalated disc, and the left-hand side of Equation (1) represents a Laplacian term in this tangent space.

As in our previously published computational models [14,16], the total ion current density was expressed as the sum (Equation (4)) of the voltage-gated Na^+^ current density (I_Na_) and a K^+^ current density (I_K_). Both ion current densities, I_Na_ (Equation (5)) and I_K_ (Equation (6)), are represented according to Ohm’s law:(4)$$I_{\mathrm{ion}}=I_{\mathrm{Na}}+I_{K}$$
(5)$$I_{\mathrm{Na}}=F_{\mathrm{ID}}\;F_{\mathrm{gNa}}\;g_{\mathrm{Na}}\;m^{3}\;h\;j\;\left(V_{m}-E_{\mathrm{Na}}\right)$$
(6)$$I_{K}=F_{\mathrm{ID}}\;g_{K}\;\left(V_{m}-E_{K}\right)$$
where g_Na_ and g_K_ represent the maximal conductances per unit area for Na^+^ and K^+^ channels, respectively. E_Na_ and E_K_ represent the respective Nernst potentials of passive ion transport. To incorporate Na^+^ channel gating, the Hodgkin and Huxley [35] formalism was used, with opening and closing rates taken from the Luo and Rudy model [36] with modifications by Livshitz and Rudy [37]. This formalism includes three gating variables (m, h, j), which were integrated using the method of Rush and Larsen [38]. Furthermore, in our modeling framework, we considered a high density of Na^+^ channels in the intercalated disc membranes (approximately 50% being located in intercalated discs [22,23,39]). Accordingly, to account for this, we used a scaling factor F_gNa_ = 5.05 as explained in detail in Hichri et al. [14] and in Ivanovic and Kucera [16]. Of note, the remaining 50% of Na^+^ channels were placed on the cylindrical lateral membrane of the cells.

A tortuous intercalated disc membrane has an increased area compared to a flat disc. If the number of channels was proportional to membrane area, this would imply a larger total number of ion channels in tortuous intercalated discs. However, the assumption that 50% of Na^+^ channels are located in the intercalated discs should hold irrespective of the intercalated disc shape. Moreover, from Hichri et al. [14], we know that the area and thus the total number of Na^+^ channels greatly influence the negative cleft potential and, hence, ephaptic coupling. Ephaptic interactions are enhanced with increasing total Na^+^ channel conductance, and the effect of Na^+^ channel density prevails over the effect of the radial resistance to the bulk [14]. Therefore, to separate the effect of convoluted structures from the effect of changing the number of Na^+^ channels on ephaptic coupling, an additional scaling factor F_ID_ was added to the right-hand sides of Equations (5) and (6). This ensured that the total Na^+^ current conductance (and K^+^ current conductance) in the intercalated disc membranes remained constant when the geometry of the ID was varied. F_ID_ was calculated as:(7)$$F_{\mathrm{ID}}=\frac{A_{\mathrm{Flat}}}{A_{\mathrm{ID}}}$$
where A_ID_ is the area of the tortuous disc and A_Flat_ is the area of the reference flat intercalated disc. For a tortuous disc, F_ID_ is always <1.

To solve the system of partial differential equations described above, we reformulated the problem as a set of ordinary differential equations. For this purpose, space was discretized with the finite element method based on triangular finite elements of piecewise linear basis functions [40,41]. For the temporal discretization, the Crank–Nicolson method was used [42]. As the initial condition of the entire intercalated disc cleft space, extracellular potential was set to zero (V_e_ = 0 mV). Figure 1A illustrates a schematic representation of the model including membrane capacitance, ion currents and extracellular resistive properties. At the rim of the disc, a Dirichlet boundary condition of zero potential was applied, assuming negligible extracellular potentials and gradients outside of the cleft (i.e., in the bulk extracellular space). For the main intracellular nodes, which were connected directly to all intracellular membrane sides, the choice of boundary conditions depended on the patch clamp protocol (voltage clamp or current clamp) being simulated. When a voltage clamp protocol was simulated (Figure 1A, left panel), a Dirichlet boundary condition was applied to the intracellular potentials (V_i,1_ and V_i,2_). For voltage clamp simulations, it was not necessary to incorporate the lateral membrane into the model. In contrast, for the simulation of a current clamp protocol (Figure 1A, right panel), a Neumann condition with an additional current source was applied to the corresponding intracellular nodes, and the bulk lateral cell membranes were incorporated. A more detailed description of boundary conditions is presented in the Section 3. All simulations were executed in MATLAB (The MathWorks, Inc., Natick, MA, USA, version R2019a) with a constant time step Δt of 0.05 µs. A detailed description of the numerical methods is presented in Hichri et al. [14].

### 2.2. Finite Element Meshes

Intercalated discs exhibit highly complex geometries where plicate and interplicate regions mingle. As documented by electron microscopy studies, interplicate regions are sometimes oriented parallel to the intercalated disc rim (corresponding to concentric folds) and sometimes oriented more radially (corresponding to radial folds) [29,30,31,33,34]. Therefore, to simulate these two types of folding patterns and to differentiate the individual effects of each pattern on ephaptic coupling, we introduced predefined geometries with concentric and radial folds, as illustrated in Figure 1B and Figure 1C, respectively. These predefined geometries were created using COMSOL Multiphysics (COMSOL AB, Stockholm, Sweden, version 5.6). Figure 1D illustrates a reference flat surface. The curved surfaces are equation-based mathematical surfaces defined using polar coordinates as functions of radius r and azimuthal angle ϕ, with r ranging from 0 to disc radius (R_ID_ = 11 µm) and ϕ from 0 to 2π.

Specifically, intercalated disc surfaces with concentric folds were defined as:(8)$$f_{C}\left(r,\phi \right)=A\cdot \cos\left(N\;\pi \;\frac{r}{R_{\mathrm{ID}}}\right)$$
where A = a R_ID_ is the amplitude of the cosine wave determining the folds, and N is the number of folds (counted from the center to the rim). A was expressed as a multiple (with multiplier a) of the intercalated disc radius (R_ID_).

Conversely, intercalated disc surfaces with radial folds were based on a parametric surface defined as:(9)$$f_{R}\left(r,\phi \right)=A\cdot \sin\left(N\;\phi \right)\;\left(\sqrt{{\left(\frac{r}{R_{\mathrm{ID}}}\right)}^{2}+q^{2}}-q\right)$$
where A determines the amplitude of the folds at the periphery, and N is the number of folds. The unitless parameter q (set to 0.2) was incorporated to avoid a singularity with sharp edges in the center of the disc. Of note, if q = 0, the expression in the last parenthesis equals r/R_ID_.

The parameters used in our model are listed in Table 1. The areas A_ID_ of the curved surfaces were computed by summing the areas of the triangular finite elements.

## 3. Results

### 3.1. Concentric Folds Potentiate Ephaptic Coupling

In the first part of our study, we focused on two simple predefined geometrical structures (shown in Figure 1B,C) representing the tortuosity of the intercalated disc. For both surface geometries (concentric and radial folds), Na^+^ and K^+^ channels were uniformly distributed in the intercalated disc membranes. Moreover, with the aim of focusing on the effect of convoluted intercalated disc structures on ephaptic coupling, gap junctional coupling was set to zero. To simulate patch clamp experiments with a cell pair, a voltage clamp protocol was applied on the intracellular side of both neighboring cells (Figure 1A, left panel). For the pre-junctional cell, the intracellular voltage protocol consisted of a depolarizing step that started at a holding potential being equal to the resting membrane potential of −85 mV. For our simulations, we considered two different voltage steps: far above the I_Na_ activation threshold (V_step_ = −25 mV) and close to the threshold (V_step_ = −50 mV). The intracellular voltage of the post-junctional cell was clamped at the resting membrane potential. Ephaptic interactions were assessed by the amplitude of the Na^+^ currents in the two membranes (obtained by summing the Na^+^ current over all finite elements) and by the temporal evolution of the minimal extracellular cleft potential.

Figure 2A shows the time course of I_Na_ in the pre- and post-junctional membranes as well as the minimal V_e_ in a series of simulations in which the amplitude of concentric folds was kept constant (A = 1/4 R_ID_) while the number of folds was varied. Due to the symmetry of the folding patterns, the minimal V_e_ occurred in the center of the intercalated disc. Thus, the minimal V_e_ curves represent the V_e_ at the central node. Our simulations show that, for both step potentials (−25 and −50 mV), concentric folds led to more negative extracellular potentials when compared with the flat intercalated disc. As expected, the V_step_ to −25 mV (far above the I_Na_ activation threshold) produced a larger peak I_Na_ in the pre-junctional membrane compared with the V_step_ to −50 mV (just above the threshold). However, for V_step_ = −50 mV, the minimal V_e_ was more negative because I_Na_ also activated in the post-junctional membrane. Furthermore, the negative V_e_ became more prominent when the number of concentric folds was increased. From our previous computational studies [14,16], we know that the negative potential in the narrow extracellular space modifies the Na^+^ current flow in the intercalated disc membranes via the mechanisms of self-activation and self-attenuation. With a step potential far above the threshold (Figure 2A, left panels), peak I_Na_ in the pre-junctional membrane decreased (in absolute value) with increasing N, which indicates increased self-attenuation. With a step potential just above the threshold (Figure 2A, right panels), peak I_Na_ in the pre-junctional membrane behaved in a more prominent manner. As highlighted by arrows in Figure 2A, self-attenuation also occurred for the pre-junctional Na^+^ current (arrow labeled “I”), but self-activation occurred as well (arrow labeled “II”), which led to a progressively earlier onset of I_Na_ with increasing N. Thus, peak I_Na_ decreased with increasing number of folds, but at the same time, the Na^+^ channels located in the intercalated disc membrane of the pre-junctional cell were activated earlier. For the post-junctional Na^+^ current, self-activation (arrow labeled “III”) occurred with increasing N. The enhancement of I_Na_ in the post-junctional membrane was caused by an increased open probability of the channels, which activated once the threshold was passed. Thus, in the voltage clamp scenario investigated here, the increase in I_Na_ was caused by the transactivation of the channels in the post-junctional membrane due to the V_e_ becoming sufficiently negative.

Figure 2B illustrates a similar simulation series in which the number of concentric folds was now kept constant (N = 4) while their amplitude was varied. The effects of increasing the amplitude of the folds were qualitatively similar to the effects of increasing their number while keeping their amplitude constant (Figure 2A). Thus, increasing either the amplitude or the number of concentric folds potentiated ephaptic effects and ephaptic coupling. The more negative cleft potential (the hallmark of ephaptic interactions) can be explained by the longer average path from a point in the cleft to the bulk extracellular space (caused by a higher amplitude and/or number of folds), which leads to an increased radial cleft resistance and thus to larger gradients of V_e_, which in turn reinforces ephaptic coupling.

It is important to note that Na^+^ current density was not homogeneous in the intercalated disc. For example, Figure 3 shows total I_Na_ as well as I_Na_ density and V_e_ at different positions in the intercalated disc with A = 1/4 R_ID_, N = 4 and a cleft width of 30 nm. For comparison, the time course of pre-junctional I_Na_ in the absence of any ephaptic coupling (free membrane) is shown in grey (Figure 3, top row). For both step potentials, −25 mV and −50 mV, the time course of I_Na_ density depended on the position in the intercalated disc (Figure 3, middle row). Specifically, I_Na_ density in the pre-junctional membrane was higher (in absolute value) close to the disc rim than in the center. For the post-junctional membrane, I_Na_ density monotonically decreased with increasing distance from the center for V_step_ = −25 mV, and it varied in a biphasic manner for V_step_ = −50 mV with a notable dispersion in time. This dispersion explains the broad time course of I_Na_ in Figure 2. Notably, for V_step_ = −50 mV, I_Na_ exhibited a radially propagated centrifugal response. Importantly, V_e_ became less negative with increasing distance from the center of the intercalated disc, where V_e_ was the most negative (Figure 3, bottom row).

The spatiotemporal pattern of I_Na_ in the intercalated disc also explains the seemingly paradoxical very small post-junctional I_Na_ for V_step_ = −25 mV. In our simulations, the post-junctional intracellular node was clamped and held at −85 mV. Because the threshold for I_Na_ to activate is near −55 mV in our model, a minimal V_e_ of at least −30 mV was required. For V_step_ = −25 mV, this threshold was hardly reached for flat and less tortuous intercalated discs (small A and/or small N, see Figure 2). Therefore, no transactivation occurred. For V_step_ = −50 mV, the situation was different because the I_Na_ built up in an asynchronous manner and during a longer period in the pre-junctional membrane, and the minimal V_e_ also built up slower. This resulted in a prolonged period near the minimal V_e_ of −30 mV corresponding to the threshold of the post-junctional membrane, which left more time for the activation gating of the post-junctional I_Na_.

In Figure 4A, to ascertain which of the parameters A and N exerts a larger influence on ephaptic coupling, simulations were run in which A and N were varied jointly such that their product remained constant (A ∙ N = 1/2 R_ID_). For a cleft width of 30 nm, all combinations of A and N led to similar effects on I_Na_ and minimal V_e_ compared to the flat intercalated disc. Thus, for concentric folds and in the ranges of A and N investigated, both parameters influenced ephaptic coupling in a similar manner in the absence of gap junctional coupling. Because cleft width is an important determinant of ephaptic coupling, we repeated these simulations with different cleft widths (from 10 to 100 nm) and examined the negative peak I_Na_ in the pre- and post-junctional membranes as well as the minimal peak V_e_ (Figure 4B). For all cleft widths, peak I_Na_ and minimal peak V_e_ were comparable for different combinations of A and N, with A ∙ N = 1/2 R_ID_. However, for tortuous intercalated discs, the minimal peak V_e_ was clearly more negative than with the reference flat intercalated disc, which also led to variations in peak I_Na_. Moreover, negative peak I_Na_ and minimal peak V_e_ were modulated by cleft width, with decreasing intermembrane spacing leading to more negative peaks of V_e_. Therefore, decreasing cleft width led to ephaptic interactions with both self-activation and self-attenuation. For V_step_ = −25 mV, the magnitude of peak I_Na_ in the pre-junctional membrane decreased slightly (in absolute value) with decreasing cleft width, reflecting self-attenuation. In contrast, in the post-junctional membrane, I_Na_ was absent for wider clefts and appeared with increasing magnitude when the cleft was narrowed below 30 nm (self-activation). For V_step_ = −50 mV, the abrupt increase in I_Na_ (in absolute value) reflected self-activation and thus ephaptic interactions in both intercalated disc membranes. However, very narrow clefts (10–20 nm) led to a less negative peak I_Na_ (self-attenuation).

### 3.2. Radial Folds Attenuate Ephaptic Coupling

We then used our computational model with the same voltage clamp protocols to study the effect of radial folds on ephaptic coupling (Figure 5). As done for concentric folds, we investigated the effects of changing the number of radial folds (Figure 5A) and changing the amplitude of folds (Figure 5B) separately from each other.

In Figure 5A, the amplitude of folds was kept constant (A = 1/4 R_ID_) while the number of folds was varied. In contrast to concentric folds, radial folds led to less negative extracellular potentials when compared with the flat intercalated disc for both step potentials (−25 and −50 mV). For a step potential far above the threshold (Figure 5A; left panels), V_e_ became slightly less negative with increasing N. However, for step potentials close to the threshold (Figure 5A; right panels), the decrease in negative V_e_ (in absolute value) was larger with increasing N. Thus, radial folds decreased ephaptic interactions, and I_Na_ in the post-junctional membrane was hardly activated. The absence of a response in the post-junctional membrane was due to the less negative V_e_, which failed to recruit a sufficient amount of I_Na_. Overall, the effect of radial folds on V_e_ was opposite to that of concentric folds and was also less intense. Nevertheless, the modulation of I_Na_ flow in the intercalated disc membranes via the mechanisms of self-activation and self-attenuation decreased for radial folds. For pre-junctional Na^+^ channels, opposite mechanisms occurred for both step potentials. For V_step_ = −25 mV, peak I_Na_ slightly increased with increasing N, reflecting self-activation and decreased self-attenuation. In contrast, for V_step_ = −50 mV, peak I_Na_ decreased (arrow labeled “I”) and was additionally delayed (arrow labeled “II”) with increasing N. Thus, for V_step_ = −50 mV, self-activation was decreased for the pre-junctional Na^+^ current by the introduction of radial folds. For the post-junctional Na^+^ current, we observed no effect on I_Na_ for V_step_ = −25 mV and a very slight decrease in the peak I_Na_ for V_step_ = −50 mV. Thus, for both step potentials, radial folds decreased self-activation for post-junctional Na^+^ channels (arrow labeled “III”).

Figure 5B shows the results of simulations in which the amplitude of folds was varied and the number of folds was kept constant (N = 4). Comparable to the results seen with concentric folds (Figure 2), increasing the amplitude or number of radial folds (Figure 5A) separately from each other led to qualitatively similar effects on the temporal evolution of the minimal V_e_ and I_Na_. Consequently, radial folds with increasing amplitude or number reduced self-activation and self-attenuation and thus diminished ephaptic effects and ephaptic coupling. This diminution can be explained by a larger number of possible paths from a given point in the cleft to the bulk extracellular space, which decreased the radial resistance in the cleft, which led to smaller gradients of V_e_, which therefore led to less ephaptic coupling.

### 3.3. Tortuous Intercalated Discs Modulate Ephaptic Coupling in a more Complex Manner in the Presence of Clustered Na^+^ Channels

In the second part of our study, we examined the effects of tortuous intercalated discs on action potential transmission from one cell to the other via ephaptic coupling. For this purpose, we utilized a current clamp protocol (Figure 1A, right panel) as follows. On the intracellular side of the pre-junctional cell, a rectangular current pulse (intensity: 11.5 nA, duration: 0.5 ms) was applied to elicit an action potential. On the intracellular side of the post-junctional cell, a current clamp protocol with zero current was applied. The intracellular potentials of both cells were recorded. As in the first part of our study, gap junctional coupling was set to zero to focus exclusively on action potential propagation via ephaptic coupling. For uniformly distributed Na^+^ channels, neither the flat nor the tortuous intercalated discs led to action potential propagation or subthreshold responses in the post-junctional cell. Action potential propagation failed because the extracellular cleft potential was not sufficiently negative. Thus, the threshold required for the Na^+^ channels on the post-junctional membrane to open was not reached.

In cardiac intercalated discs, it has been shown that Na^+^ channels are not uniformly distributed but instead form clusters [12,23,24,25]. The computational study by Hichri et al. [14] showed that action potential propagation via ephaptic coupling is facilitated when Na^+^ channels are clustered in the center of the intercalated disc. Therefore, we redistributed the conductance of the Na^+^ current of the two intercalated disc membranes into centered circular clusters with a radius R_c_ = 1/8 R_ID_. Figure 6 and Figure 7 show the time course of the intracellular potentials (V_i_), I_Na_ and minimal V_e_ for concentric folds (Figure 6) and radial folds (Figure 7), respectively.

Figure 6A illustrates a simulation series in which we varied the number and amplitude of concentric folds separately from each other (as in Figure 2). The cleft width was set to 30 nm over the entire intercalated disc. For concentric folds, the minimal V_e_ in the cleft became more negative with increasing N when A was kept constant at 1/4 R_ID_ (Figure 6A; upper panels). Due to the more negative cleft potential V_e_ (up to −120 mV for N = 8), I_Na_ was strongly modulated by N. For the pre-junctional membrane, peak I_Na_ decreased and was delayed with increasing N, indicating increased self-attenuation. For the post-junctional membrane, increasing N also led to increased self-attenuation, as reflected by a decreased and delayed peak I_Na_. Of note, in line with our previous work [16], the pre-junctional I_Na_ switched from an inward to an outward current. Hence, it provided positive charges to the cleft. For the specific case of I_Na_, although the ion concentrations were constant in our model, we surmise that this outward I_Na_ provides the cleft with Na^+^ ions, which can then enter the post-junctional cell via activated Na^+^ channels (a hypothesis which we termed “Na^+^ transfer” [16]). However, with increasing N, less Na^+^ transfer occurred. Regarding the intracellular potential of the post-junctional cell, contrasting responses resulted from the excitation of the pre-junctional cell membrane. With the flat intercalated disc, the action potential propagated from the pre- to the post-junctional cell. In contrast, for tortuous intercalated discs, the threshold required to depolarize the post-junctional cell was not reached. Thus, for concentric folds with A = 1/4 R_ID_ and N between 1 and 4, only subthreshold depolarization occurred. Moreover, V_i_ shifted further away from the threshold with increasing N. The lower panels of Figure 6A show that the effects of increasing the amplitude of concentric folds while keeping their number constant (N = 4) were qualitatively comparable. Thus, increasing either the number or the amplitude of concentric folds potentiated ephaptic interactions. These results are in line with those observed with uniformly distributed Na^+^ channels (Figure 2 and Figure 4).

Figure 6B shows the same simulation series for concentric folds as in Figure 6A, but this time the cleft width was set to 40 nm. Again, with increasing N (Figure 6B; upper panels) and A (Figure 6B; lower panels) the minimal V_e_ became more negative (up to −120 mV for N = 8 or A = 1/2 R_ID_, respectively), which caused a large modulation of I_Na_. As for a cleft width of 30 nm (Figure 6A), the separate variation of N and A led to comparable simulation results. For pre-junctional I_Na_, increasing N or A resulted in self-attenuation with a later onset and decreased peak I_Na._ However, for post-junctional I_Na_, self-activation and self-attenuation were modulated in a biphasic manner when N or A were varied. Compared with the flat intercalated disc, peak I_Na_ increased (in absolute value) and the onset of I_Na_ was delayed. However, peak I_Na_ did not monotonically increase with increasing N or A. Specifically, post-junctional I_Na_ occurred earlier when N was increased from 1 to 2 (Figure 6B; upper panels) or when A was increased from 1/16 R_ID_ to 1/8 R_ID_ (Figure 6B; lower panels), and it then occurred later when N and A were further increased. Moreover, the inward I_Na_ of the pre-junctional membrane switched to an outward I_Na_ for tortuous intercalated discs but not for the flat disc. Regarding the intracellular potential of the post-junctional cell, no manifest depolarization occurred with the flat intercalated disc with an increased intermembrane spacing of 40 nm. In contrast, for N = 1 (Figure 6B; upper panels) and A = 1/16 R_ID_ (Figure 6B; lower panels), the action potential propagated from the pre- to the post-junctional cell. Nevertheless, further increase in either N or A led to subthreshold depolarization. Thus, in the presence of centered Na^+^ channel clusters, concentric folds potentiated ephaptic coupling due to increased radial cleft resistance leading to very negative extracellular potentials. However, as self-attenuation increased for post-junctional I_Na_, action potential propagation depended in a more complex manner on cleft width and the number and amplitude of concentric folds.

In Figure 7, a similar simulation series was conducted as in Figure 6, but this time the effects of radial folds were investigated in the presence of a central Na^+^ channel cluster. Figure 7A illustrates this simulation series for a cleft width of 30 nm. For N up to 4 (Figure 7A; upper panels; N was varied while A was kept constant) or A up to 1/4 R_ID_ (Figure 7A; lower panels; A was varied while N was kept constant), the minimal V_e_ in tortuous intercalated discs was similar to the reference minimal V_e_ in a flat disc. With an increased number (N = 8, top row) or amplitude (A = 1/2 R_ID_, bottom row) of radial folds, V_e_ became about half as negative as in the flat intercalated disc. This goes in line with the almost complete absence of I_Na_ in the post-junctional membrane and the absence of depolarization in the post-junctional intracellular node. Moreover, compared with concentric folds (Figure 6), radial folds modulated I_Na_ in the intercalated disc in a different manner via the mechanisms of self-activation and self-attenuation. For the pre-junctional membrane, peak I_Na_ increased with larger N or A (self-activation and less self-attenuation), while for the post-junctional membrane, peak I_Na_ decreased (self-attenuation and less self-activation). Furthermore, the outward component of pre-junctional I_Na_ decreased with increasing N and A and remained inward for N = 8 (Figure 7A; top row) or A = 1/2 R_ID_ (Figure 7A; bottom row). Regarding V_i_, the delay between the action potential upstrokes shortened with increasing N and A compared with the flat intercalated disc until, for very tortuous intercalated discs (e.g., N = 8 or A = 1/2 R_ID_), V_i_ remained close to resting membrane potential, thus resulting in no action potential propagation. Figure 7B illustrates the same simulation series for radial folds as in Figure 7A but with an intermembrane spacing of 40 nm. With the flat intercalated disc, the minimal V_e_ was half as negative as compared with a cleft width of 30 nm. Thus, in this scenario, increasing N (Figure 7B; upper panels) or A (Figure 7B; lower panels) led to a very slightly less negative V_e_, which modulated I_Na_ only minimally. Similar to a cleft of 30 nm (Figure 7A), increasing N or A decreased self-attenuation for pre-junctional I_Na_ and decreased self-activation for post-junctional I_Na_. In contrast, the inward I_Na_ of the pre-junctional membrane did not switch to an outward I_Na_. Moreover, no manifest depolarization occurred in the post-junctional cell, neither with the flat nor with tortuous intercalated discs. Thus, in line with the simulation results observed for uniformly distributed Na^+^ channels (Figure 5), radial folds decreased the radial resistance of the cleft, leading to less negative extracellular cleft potentials and attenuated ephaptic effects also in the presence of clustered Na^+^ channels.

For concentric (Figure 6) as well as radial folds (Figure 7), the response of the post-junctional cell was modulated by the intermembrane spacing of two opposing intercalated discs. Therefore, in Figure 8A, we summarize the three different responses (V_i_) of the post-junctional cell (no depolarization, subthreshold depolarization or action potential propagation) as a function of the product of A and N and the cleft width. We used the product of A and N because qualitatively similar simulation results were obtained when both parameters were varied separately from each other (see Figure 6 and Figure 7). Nevertheless, some combinations of A and N with a constant product led to different responses (Figure 8A; double-colored dots). For concentric folds (Figure 8A; right panel) and a very narrow cleft width of 20 nm, all tested combinations of A and N led to subthreshold depolarization, whereby Na^+^ channels in the intercalated disc membrane activated, but the required threshold to depolarize the post-junctional cell was not reached. When the cleft width was increased to 30 nm, subthreshold depolarization occurred for intercalated discs with larger tortuosity (A ∙ N ≥ 1/4 R_ID_). However, for the flat intercalated disc and discs with lower tortuosity (A ∙ N ≤ 1/8 R_ID_), the action potential propagated from the pre- to the post-junctional cell. With even wider clefts, the action potential propagated from one cell to the next with increasing product of A and N, but only for a narrow range of tortuosity (e.g., A ∙ N = 1/4 R_ID_ for a cleft width of 40 nm and A ∙ N = 1/2 R_ID_ for a cleft width of 50 nm). A larger product outside of this range led again to subthreshold depolarization, while for smaller products of A and N, no manifest depolarization occurred in the post-junctional cell. For cleft widths above 70 nm, no depolarization was observed regardless of the amplitude and number of concentric folds. In comparison, no depolarization occurred with radial folds (Figure 8A; left panel) when cleft widths exceeded 30 nm. However, for a cleft width of 30 nm, the action potential propagated for most of the combinations of A and N (A ∙ N ≤ 1 R_ID_), which was also observed for the flat intercalated disc membrane. For a narrower cleft width of 20 nm, action potential propagation only occurred for A ∙ N = 2 R_ID_. Overall, Figure 8A shows that in the presence of Na^+^ channel clusters, V_i_ was mostly modulated by cleft width rather than by the amplitude and number of radial folds. However, for concentric folds, action potential propagation was facilitated via enhanced ephaptic coupling at cleft widths of 40 and 50 nm. On the one hand, for wide clefts, action potential transmission failed because the radial resistance towards the bulk was too low, precluding self-activation. On the other hand, for narrow clefts, self-attenuation reduced post-junctional I_Na_, leading to insufficient depolarization of the bulk membrane of the post-junctional cell (resulting in only a subthreshold response). For narrow clefts or very tortuous intercalated discs with concentric folds, the action potential was not transmitted due to insufficient self-activation (transactivation) in the post-junctional membrane. Moreover, for radial folds, the decreased radial resistance to the bulk rendered ephaptic coupling and transactivation more sensitive to cleft width.

In Figure 8B, the delay between the action potential upstrokes of the pre- and post-junctional cells is illustrated for a cleft width of 30 nm and different combinations of A and N leading to action potential propagation. The delay was prolonged for concentric folds (Figure 8B; right panels) but shortened for radial folds (Figure 8B; left panels) when compared with the flat intercalated disc and for increasing A and N. This opposite effect on the delay can be mostly explained by differences in the minimal V_e_ and modulation of I_Na_, which is in line with the results in Figure 6 and Figure 7. Thus, in the presence of clustered Na^+^ channels in the center of intercalated discs, concentric folds potentiated ephaptic effects while radial folds attenuated them. These results are also in line with the first part of our study, where Na^+^ channels were uniformly distributed in the intercalated discs. However, a very negative cleft potential V_e_ can lead to increased self-attenuation of post-junctional I_Na_. Consequently, whether action potential propagation via ephaptic coupling is facilitated is highly dependent on Na^+^ channel clustering and cleft width as well as on the orientation, number and amplitude of geometrical folds.

## 4. Discussion

### 4.1. Main Results

To date, providing direct experimental evidence of ephaptic coupling is difficult because it is hardly possible to measure the extracellular potential in nanoscale clefts. Modeling studies thus provide valuable insights. Using our refined finite element model of an intercalated disc shared by two cardiomyocytes, we investigated the effect of tortuous intercalated discs on ephaptic coupling. We tested predefined folding patterns parametrized by the orientation (concentric vs. radial), the amplitude and the number of folds. Our main finding is that different orientations of folds have an opposite and asymmetric effect on the extracellular potential (V_e_) in the cleft and thus on I_Na_ flow in the intercalated disc membranes. This effect is essentially modulated by the radial cleft resistance to the bulk interstitium. When Na^+^ channels were uniformly distributed in intercalated discs (uniform I_Na_ conductance), ephaptic mechanisms were potentiated by concentric folds but slightly attenuated by radial folds. For both orientations, the effect on V_e_ and I_Na_ became more prominent with increasing number and amplitude of folds. Furthermore, both number and amplitude of folds modulated V_e_ and I_Na_ in a similar manner. Another main finding is that when Na^+^ channels were clustered in the center of the intercalated discs, ephaptic mechanisms were also potentiated by concentric folds and attenuated by radial folds. However, concentric and radial folds then modulated V_e_ and I_Na_ in a more complex manner. For instance, the inward Na^+^ current in the pre-junctional membrane changed its direction to an outward current, which then possibly provides the cleft with Na^+^ ions for the inward Na^+^ current through the post-junctional intercalated disc membrane. This phenomenon was already observed and termed “Na^+^ transfer” in our previous study [16], where narrow perinexi were confirmed to be privileged sites for ephaptic coupling. In the present work, concentric and radial folds modulated this outward phase of I_Na_ in different manners. Importantly, it must be noted that Na^+^ transfer relies on the assumption that Na^+^ channel clusters face each other across the cleft, which has to date not been demonstrated by morphological studies. However, it has been proposed that in perinexi adjacent to gap junctions, Na^+^ channels are arranged face-to-face across the cleft by the adhesion function of their ß1 subunits [19].

### 4.2. Physiological and Translational Relevance

The intercalated disc, with its macromolecular complexes and greatly intricate geometrical structure, ensures cardiac conduction. However, pathological structural changes in intercalated discs are known to alter conduction and are therefore associated with a higher risk of arrhythmias [19,43,44,45,46,47,48,49]. For instance, intercalated discs in older mice as well as in mice suffering from dilated cardiomyopathy were characterized by a higher degree of membrane convolution when compared with wild-type animals [43]. Moreover, in an ovine tachypacing-induced heart failure model, Pinali et al. [26] observed remodeling of the intercalated disc structure, including increased amplitude of plicate folds, increased length of interplicate regions and increased surface area. Similar changes in the intercalated disc structure were observed in αT-catenin knockout mice [32], and intercalated discs of plakophilin-2 deficient mice were also characterized by increased surface area [23].

Previous modeling studies focused on the distribution of Na^+^ channels [14] and gap junctions [16,34] in intercalated discs with homogeneous [14] or heterogeneous intermembrane separation [16,34] between adjacent cells. In the present study, we focused on intercalated disc tortuosity, hence contributing to an important investigation of the factors influencing ephaptic coupling and thus cardiac conduction. Understanding the electrophysiological roles of the geometrical structure of intercalated discs in combination with their macromolecular assemblies may, in the future, lead to new potential targets to prevent and treat cardiac arrhythmias.

### 4.3. Limitations and Perspectives

To keep the computational effort tractable, some simplifications were applied to the modeling framework. Instead of using our full three-dimensional framework of two adjacent cardiomyocytes [16], we based our study on the two-dimensional model of a shared intercalated disc [14]. In this model, the intercellular domains are reduced to one single node per cell, which accordingly simplifies and reduces the representation of the intracellular domains [14]. Consequently, we did not investigate the distribution of gap junctional plaques in tortuous intercalated discs. However, we previously showed that ephaptic interactions are only minimally modulated in the presence of a gap junction plaque in the absence of a narrowed perinexus [16]. Importantly, in our model, intercalated discs were represented in high-resolution, and the incorporation of tortuous surface meshes increased the number of finite element nodes compared to a mesh of a flat intercalated disc. Thus, the two-dimensional modeling framework refined by a third spatial coordinate represented the best choice for our study, as it led to focused simulations with the advantage of amenable computational effort.

To exclusively focus on the geometrical aspect of cardiac intercalated discs, we removed further confounding factors by applying additional simplifications to our model. In addition to the absence of gap junctional plaques, K^+^ channels were always uniformly distributed and Na^+^ channels were either uniformly distributed or regrouped in one single cluster in the center of the intercalated discs. Although regrouping all Na^+^ channels in one single central cluster is an extreme and unrealistic scenario (and thus a major limitation), we believe that it is a necessary and an important step to gaining mechanistic understanding of action potential propagation via ephaptic coupling, as we have previously done [14,16]. Gaining further understanding of what is happening with several clusters will require very careful evaluation and interpretation, starting, for example, with how two clusters interact and influence each other as a function of their respective positions. The interpretation will be even more challenging if three or more clusters are present. Nevertheless, we believe that the mechanistic understanding obtained with one central Na^+^ channel cluster will be applicable to more realistic intercalated discs incorporating several clusters. We surmise that Na^+^ channel clusters located closer to the disc periphery will lead to smaller ephaptic effects than clusters located in the center. We nevertheless note that the simulations with uniform Na^+^ channel distribution represented the limiting case of a large number of small clusters distributed uniformly. From this viewpoint, our simulations considered two limiting cases.

In terms of ion currents, only the Na^+^ current and a linear K^+^ current were incorporated, while further currents, such as the L-type calcium current, were not considered in our modeling framework. Using a linear K^+^ current rather than one more closely resembling the inwardly rectifying I_K1_ current might also affect the results since depolarization will reduce I_K1_. In this regard, we expect that the reduction of I_K1_ with depolarization will support ephaptic action potential transmission by causing a smaller repolarizing shunt current.

A further important limitation is that we assumed ion concentrations to be constant in space and time, although these are likely to vary in the intercalated disc cleft [8,34,50,51]. Ion concentration changes (and possibly also the associated change of single channel conductance) will affect ephaptic mechanisms, particularly Na^+^ transfer. To incorporate dynamic ion concentration changes into our computational model, it will be necessary to describe ion fluxes using the Nernst–Planck equation of electrodiffusion rather than Ohm’s law [8,52]. In the Nernst–Planck formalism, all ions (including anions) contribute to current density in the intra- and extracellular spaces. Thus, both cations (e.g., Na^+^, K^+^ and Ca^2+^) and anions (e.g., Cl^−^ and HCO_3_^−^) must be considered to model ion concentration changes. With respect to the Na^+^ transfer mechanism, it is possible that our simulations underestimated the outward I_Na_ because the Nernst potential for Na^+^ was kept constant instead of becoming less positive as the cleft is depleted of Na^+^ and as Na^+^ accumulates in the subsarcolemmal space. Experimentally, it was shown in Langendorff-perfused guinea pig ventricles that altering extracellular Na^+^ and Ca^2+^ concentrations modulates conduction velocity via ephaptic coupling [53,54]. Specifically, decreasing perinexal width accelerated conduction in the presence of a normal Na^+^ concentration, but it slowed conduction when the Na^+^ concentration was decreased in the perfusate [53]. These results therefore suggest that the relationship between excitability, conduction velocity, Na^+^ concentration and perinexal width is surprisingly complex [54]. Incorporating ion concentration changes into our model according to the Nernst–Planck formalism may therefore provide an explanation for these intriguing findings in the future.

Another major limitation of our study is the representation of the intercalated disc by controlled, simple, predefined folding patterns. Additionally, intermembrane spacing was kept uniform over the entire intercalated disc. As shown by microscopy studies [23,26,27,28,29,30,31,32,34], the intercalated disc is a tremendously complex, irregular and asymmetric structure with intricately mingled plicate and interplicate regions with heterogeneous cleft width. To estimate the effects of radial resistance in realistic discs would require dedicated analyses based on a large number of irregular intercalated disc structures. The reason for this is that the radial cleft resistance then depends on direction, thereby complicating the interpretation of its effects on ephaptic coupling. Hence, to gain mechanistic insight, we preferred to focus on regular folded structures with homogeneous intermembrane spacing. Nevertheless, our approach of using simplified but predefined folding geometries offered the advantage of untangling the effects of the orientation of folds. The relatively simple, but easily controllable, meshes permitted the isolated investigation of intercalated disc geometries on underlying mechanisms of ephaptic coupling.

The structural complexity of the intercalated disc motivated Moise et al. [34] to develop an algorithm that generates more realistic meshes of intercalated discs based on measurements from transmission electron microscopy images. Moise et al. [34] investigated the effects of heterogeneous intermembrane distances on conduction velocity in a cell fiber model. They showed that a larger intermembrane distance within interplicate regions increased conduction velocity for low levels of gap junctional coupling, while it reduced conduction velocity for high levels of gap junctional coupling [34]. Additionally, narrower interplicate intermembrane separation led to a less sensitive relationship between conduction velocity and gap junctional coupling [34]. Because the incorporation of these high-resolution meshes into their single cell fiber model would lead to very high computational expenses, Moise et al. [34] reduced their meshes to a smaller number of nodes at the cost of losing the detailed intercalated disc representation.

It is important to understand that ephaptic coupling is complex and depends on multiple factors, such as Na^+^ channel distribution, cleft width and tortuosity. We believe that investigating the effects of spatial features at the level of the single disc, which could be done by integrating these very high-resolution realistic meshes [34] into our intercalated disc model (including a more realistic distribution of gap junctions, Na^+^ channels and K^+^ channels), is worth exploring in the future and might lead to further insights regarding the modulation of ephaptic coupling by intercalated disc tortuosity and macromolecular complexes. Investigating the consequences of ion concentration changes, which are likely to occur in intercalated disc nanospaces, is worthy of further investigation as well. While all these highly relevant aspects are on our agenda, they were outside the scope of the present study. Here, our primary goal was to explore the effect of disc geometry and not to develop the most sophisticated model, which would be characterized by a large number of variables (cleft geometries, cluster distributions and positions and cleft widths) and would therefore generate results difficult to interpret. Extending the intercalated disc modeling framework in this direction will require more substantial computational resources and efforts with large numbers of individual simulations.

## 5. Conclusions

Our results indicate that the folding pattern of the intercalated disc, in addition to the cleft width and the clustering of Na^+^ channels, plays an important role in modulating action potential propagation via ephaptic coupling. The intercalated disc has a highly organized but complex structure, and understanding the mechanisms of ephaptic coupling requires step-by-step progress. This study represents a first and important step towards a more comprehensive understanding of how intercalated disc tortuosity modulates ephaptic coupling. It is in our future plans to establish a comprehensive model. However, besides being a technical achievement, the development of a complete model would per se not provide insights without extensive investigations. Indeed, understanding all factors that influence ephaptic coupling from a mechanistic point of view will require a careful dissection of the problem, i.e., the investigation of individual factors separately and then in combination. Hence, we are convinced that such an approach will permit a more complete investigation of the structure–function relationship of the intercalated disc and its role in cardiac excitation at the cellular level.

## Figures and Tables

**Figure 1 cells-11-03477-f001:**
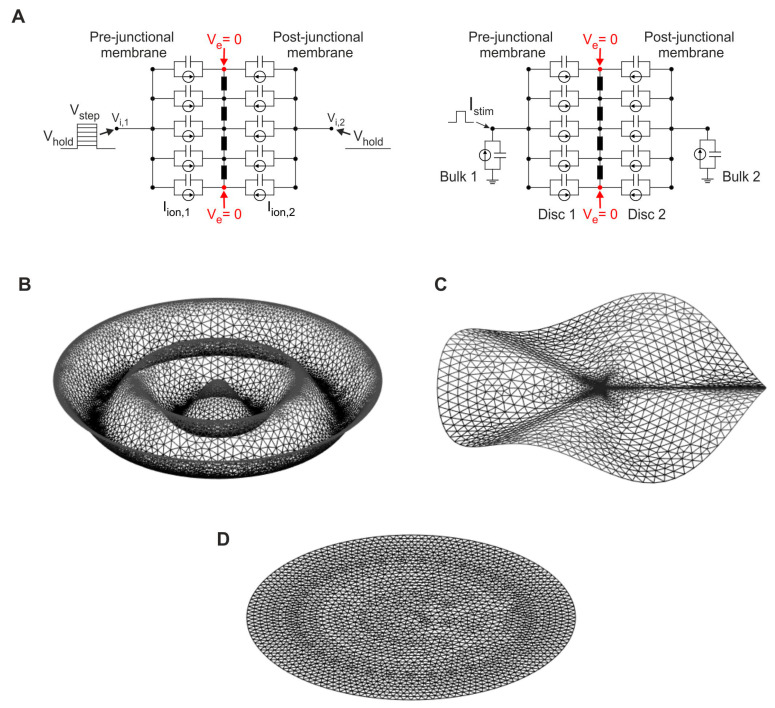
Different finite element geometries and meshes. (**A**) Schematic representation of the model for voltage clamp (left) and current clamp protocols (right). The extracellular potential (V_e_) at the rim of the disc was set to zero (red). The schematic was modified from Hichri et al. [14]. (**B**) Intercalated disc mesh consisting of concentric folds parametrized by amplitude A and number of folds N. In this example, A = 1/8 R_ID_ and N = 4. (**C**) Intercalated disc mesh consisting of radial folds parametrized by A and N. In this example, A = 1/8 R_ID_ and N = 4. (**D**) Reference mesh of a flat intercalated disc.

**Figure 2 cells-11-03477-f002:**
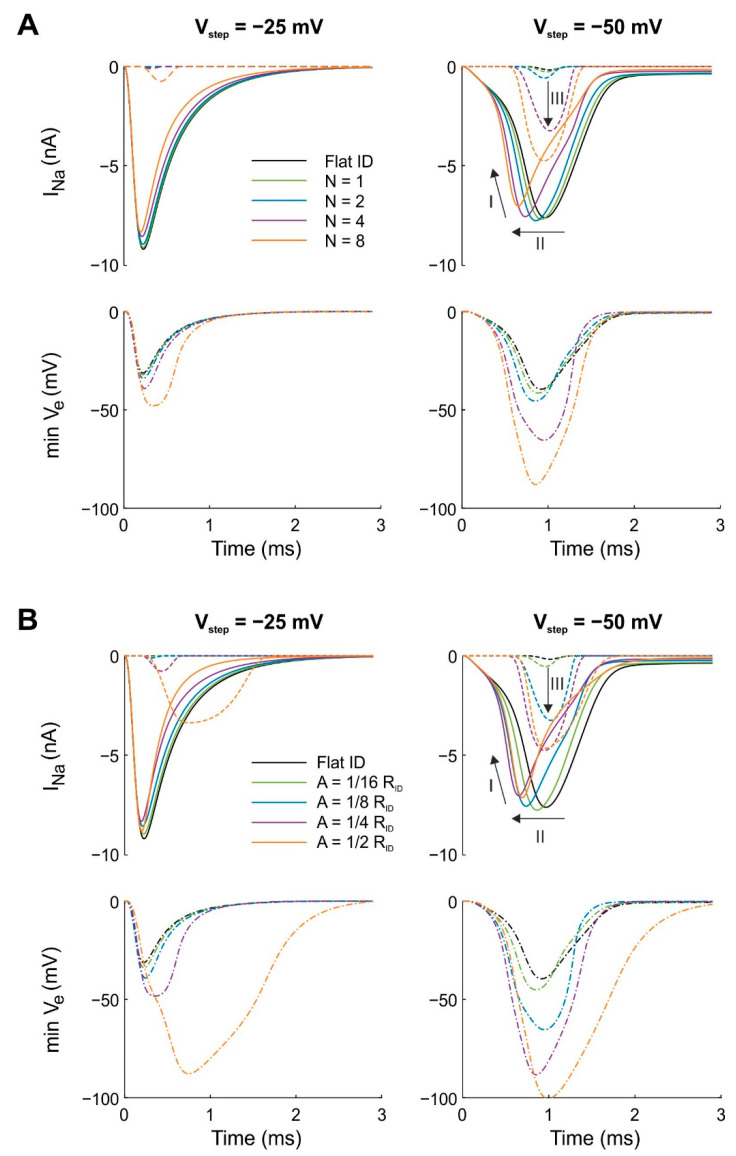
Effects of the number of folds N and the amplitude of folds A for concentric folds on I_Na_ and minimal V_e_ for a uniform distribution of g_Na_. The intracellular potential of the pre-junctional cell was subjected to a voltage clamp protocol with V_step_ = −25 mV (left) and V_step_ = −50 mV (right) while the intracellular potential of the post-junctional cell was clamped at resting membrane potential. The cleft width was set to 30 nm and gap junctional coupling was set to zero. Continuous and dotted curves correspond to I_Na_ in the pre- and post-junctional membranes, respectively. Dashed-dotted lines represent minimal V_e_. Reference simulations with a flat intercalated disc (ID) are shown in black. (**A**) The amplitude of the folds was kept constant (A = 1/4 R_ID_) while the number of folds was varied from 1 to 4. (**B**) The number of folds was kept constant (N = 4) while their amplitude was varied from 1/16 R_ID_ to 1/2 R_ID_. Arrows: I: self-attenuation of I_Na_ in the pre-junctional membrane; II: self-activation in the pre-junctional membrane; III: self-activation in the post-junctional membrane.

**Figure 3 cells-11-03477-f003:**
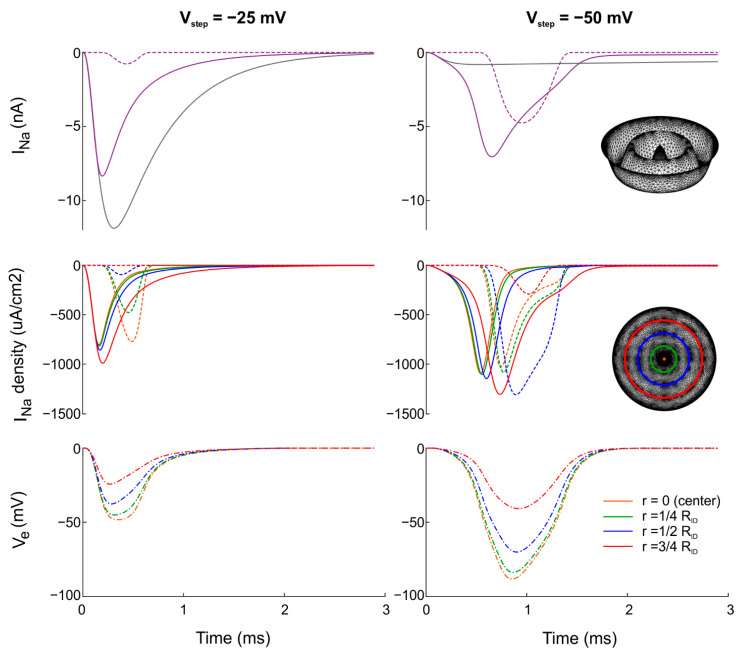
Time course of total I_Na_ with I_Na_ density and V_e_ at different positions in the intercalated disc for a uniform distribution of g_Na_. Intercalated disc with concentric folds (A = 1/4 R_ID_ and N = 4). The cleft width was 30 nm. The intracellular nodes were subjected to a voltage clamp protocol with V_step_ = −25 mV (left) and V_step_ = −50 mV (right). Top: total I_Na_ in the tortuous intercalated disc (purple). For comparison, total I_Na_ in the pre-junctional membrane without any ephaptic coupling (free membrane) is shown in grey. Middle: I_Na_ density at different distances from the center of the intercalated disc (inset). Continuous and dotted curves correspond to total I_Na_ and I_Na_ density in the pre- and post-junctional membranes, respectively. Bottom: V_e_ at the corresponding positions in the intercalated disc.

**Figure 4 cells-11-03477-f004:**
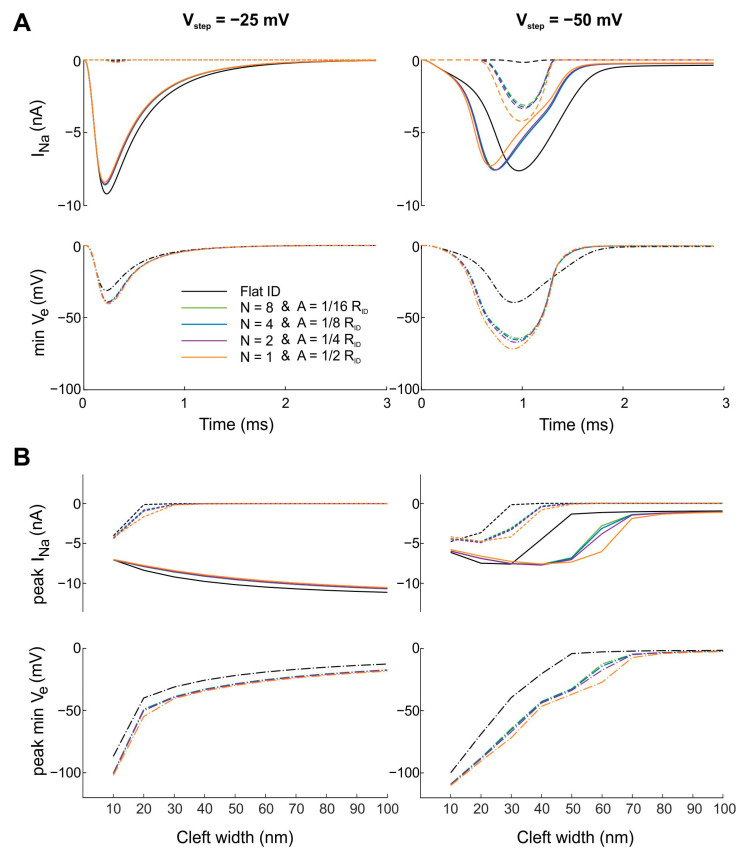
Effects of jointly changing the number N and amplitude A of concentric folds such that their product remains constant (A ∙ N = 1/2 R_ID_) on ephaptic coupling (uniform distribution of g_Na_). The intracellular nodes were subjected to a voltage clamp protocol with V_step_ = −25 mV (left) and V_step_ = −50 mV (right). (**A**) The cleft width was set to 30 nm. Continuous and dotted curves correspond to I_Na_ in the pre- and post-junctional membranes, respectively. Reference simulations with a flat intercalated disc (ID) are shown in black. (**B**) Peak I_Na_ and peak minimal V_e_ for the same combinations of A and N (color coding as in A) when cleft width was varied from 10 to 100 nm in steps of 10 nm.

**Figure 5 cells-11-03477-f005:**
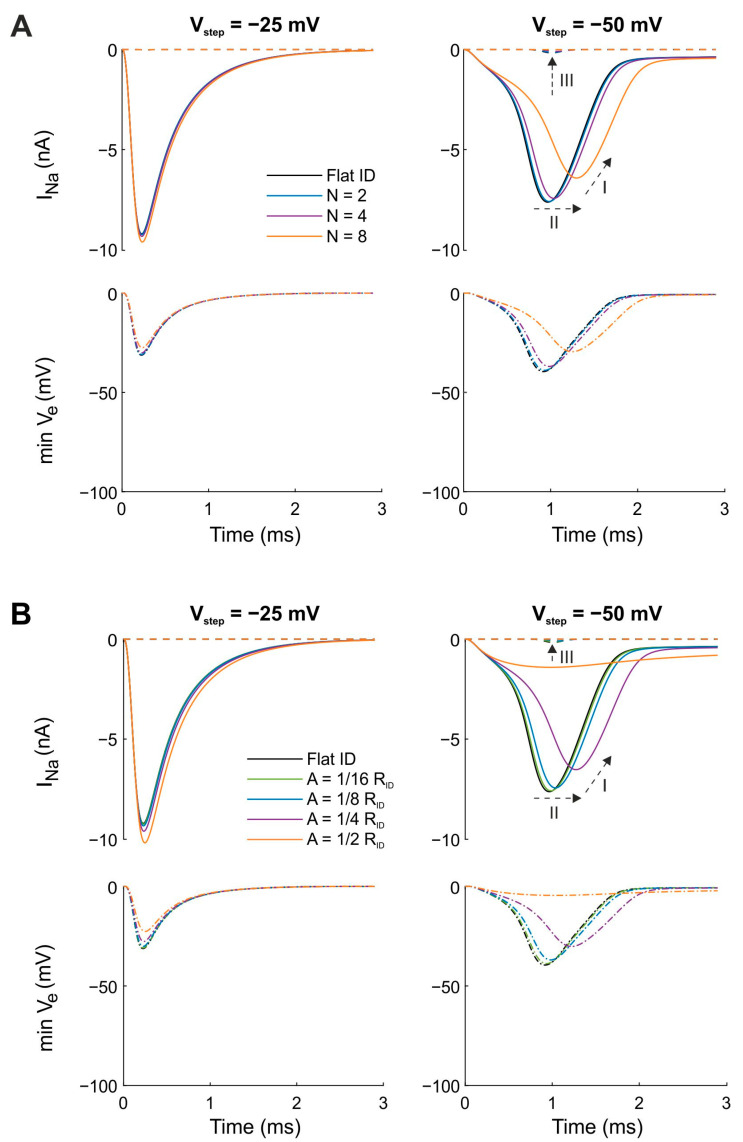
Effects of changing the number N and amplitude A of radial folds on I_Na_ and minimal V_e_ for a uniform distribution of g_Na_. Contains the same simulations and figure layout as in Figure 2 but now shows results for an intercalated disc mesh with radial folds. (**A**) The amplitude of the folds was kept constant (A = 1/4 R_ID_) while the number of folds was varied from 2 to 8. (**B**) The number of folds was kept constant (N = 4) while their amplitude was varied from 1/16 R_ID_ to 1/2 R_ID_. Dotted arrows: I: self-attenuation of I_Na_ in the pre-junctional membrane; II: self-attenuation and less self-activation in the pre-junctional membrane; III: less self-activation in the post-junctional membrane.

**Figure 6 cells-11-03477-f006:**
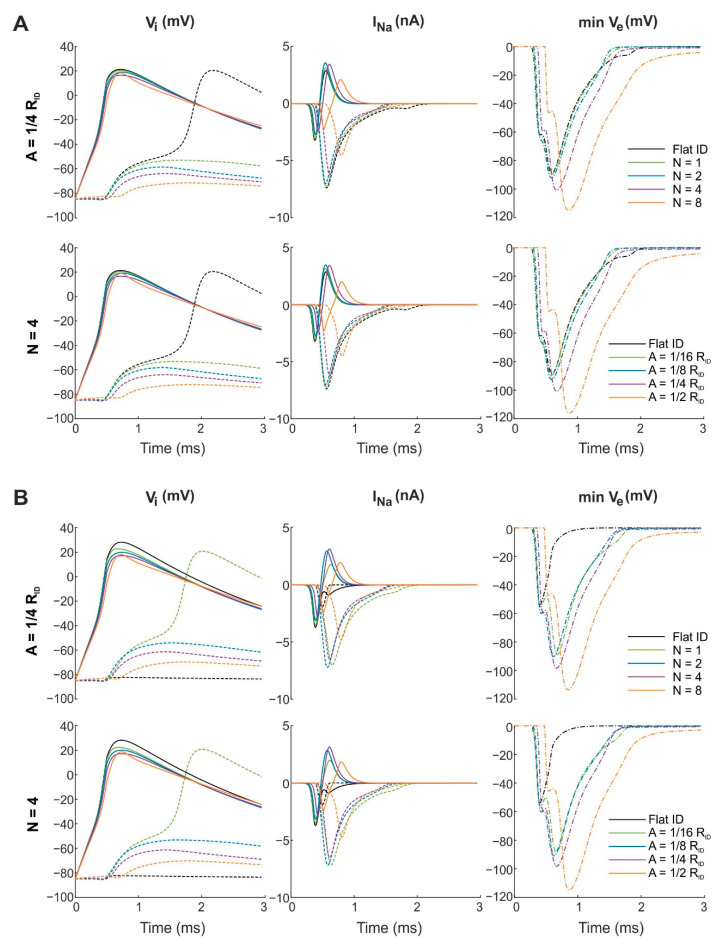
Effects of changing the number N and amplitude A of concentric folds on V_i_, I_Na_ and minimal V_e_ with Na^+^ channels clustered in the center of the intercalated disc (R_c_ = 1/8 R_ID_). The intracellular node of the pre-junctional cell was subjected to a current clamp protocol with a rectangular current pulse (intensity: 11.5 nA; duration: 0.5 ms). On the intracellular node of the post-junctional cell, a current clamp protocol with zero current was applied. Gap junctional coupling was set to zero. Continuous and dotted curves correspond to I_Na_ in the pre- and post-junctional membranes, respectively. Reference simulations with a flat intercalated disc (ID) are shown in black. (**A**) The cleft width was set to 30 nm. First row: The amplitude of the folds was kept constant (A = 1/4 R_ID_) while the number of folds N was varied from 1 to 4. Second row: The number of folds was kept constant (N = 4) while their amplitude A was varied from 1/16 R_ID_ to 1/2 R_ID_. (**B**) The cleft width was set to 40 nm. Number and amplitude of folds were varied as in panel **A**.

**Figure 7 cells-11-03477-f007:**
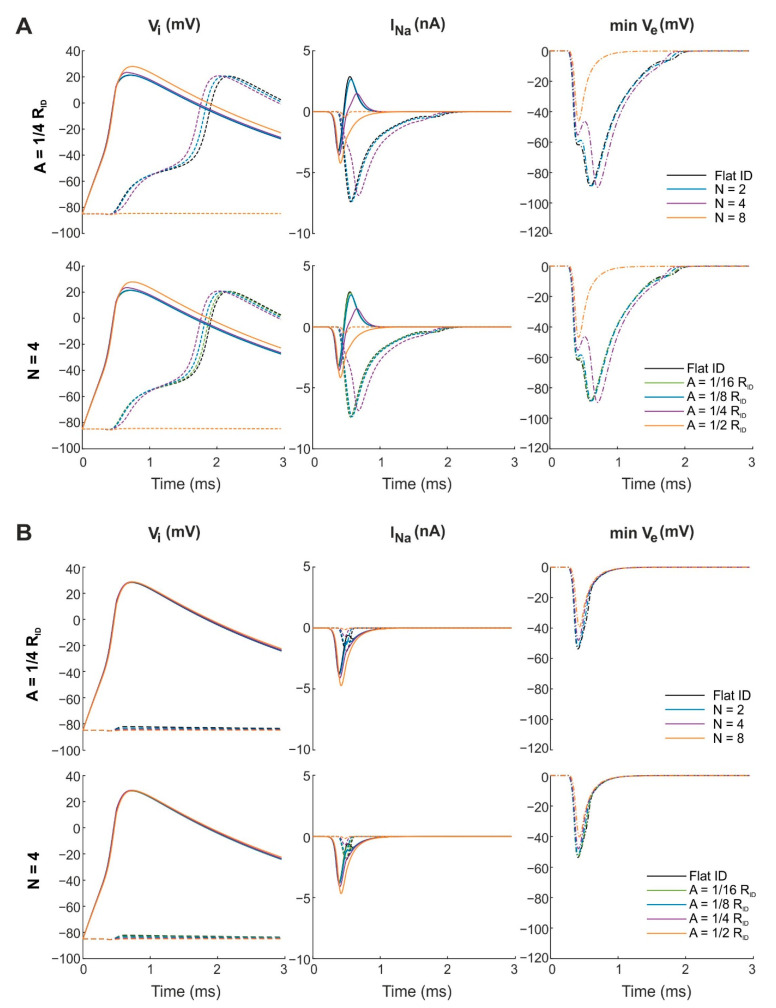
Effects of changing the number N and amplitude A of radial folds on V_i_, I_Na_ and minimal V_e_ with Na^+^ channels clustered in the center of the intercalated disc (R_c_ = 1/8 R_ID_). Contains the same simulations and figure layout as in Figure 6 but now shows results for an intercalated disc mesh with radial folds. (**A**) The cleft width was set to 30 nm. First row: The amplitude of the folds was kept constant (A = 1/4 R_ID_) while the number of folds N was varied from 2 to 4. Second row: The number of folds was kept constant (N = 4) while their amplitude A was varied from 1/16 R_ID_ to 1/2 R_ID_. (**B**) The cleft width was set to 40 nm. Number and amplitude of folds were varied as in panel **A**.

**Figure 8 cells-11-03477-f008:**
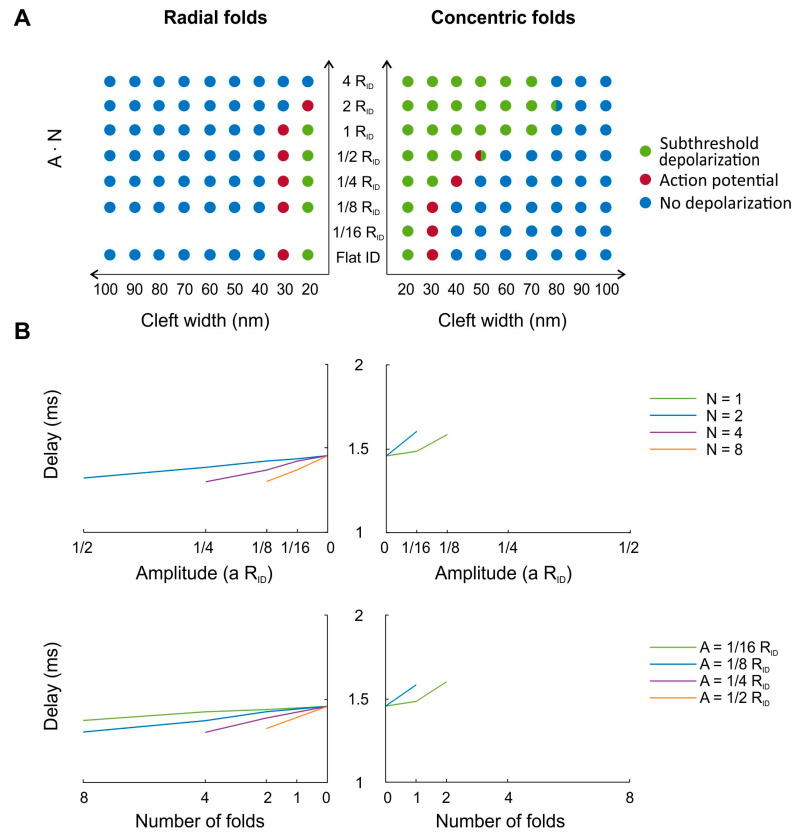
Comparison of the effects of concentric and radial folds in the presence of clustered Na+ channels in the center of the intercalated disc (Rc = 1/8 R_ID_). (**A**) Response of the post-junctional cell for radial (left) and concentric folds (right) and for different combinations of cleft width and tortuosity (i.e., value of A ∙ N). The responses were categorized as subthreshold depolarization (green), action potential (red) or no depolarization (blue). Note that for some combinations of A and N, two different responses occurred (double-colored dots). Note also that for radial folds, simulations with A ∙ N = 1/16 R_ID_ were not run because this involves simulations with N = 1 (which corresponds to an oblique but flat intercalated disc). (**B**) Delay (measured at Vi = 0 mV) between the action potential upstrokes of the pre- and post-junctional cells for radial (left) and concentric folds (right) as a function of A and N. The cleft width was set to 30 nm. The zero on the abscissas correspond to the flat intercalated disc.

**Table 1 cells-11-03477-t001:** Electrical and geometrical model parameters including symbol, definition, value and unit.

Symbol	Definition	Value	Unit
C_m_	Membrane capacitance	1	µF/cm^2^
E_Na_	Nernst potential for Na^+^	55	mV
E_K_	Nernst potential for K^+^	−85	mV
g_Na_	Maximal conductance for Na^+^	23	mS/cm^2^
g_K_	Maximal conductance for K^+^	0.3	mS/cm^2^
σ_e_	Extracellular conductivity	6.666	mS/cm
w	Cleft width	varied	nm
F_gNa_	Scaling factor for Na^+^ channel density on flat intercalated disc membranes	5.05	unitless
F_ID_	Scaling factor for Na^+^ channel density on tortuous intercalated disc membranes	varied	unitless
A_ID_	Area of the tortuous intercalated disc	varied	µm^2^
A_Flat_	Area of the reference flat intercalated disc	$380.13\;\left({\pi \;R}_{\mathrm{ID}}^{2}\right)$	µm^2^
R_ID_	Radius of the intercalated disc	11	µm
A	Amplitude (peak magnitude) of folding	varied	µm
a	Multiplier of R_ID_	varied	unitless
N	Number of folds	varied	unitless
q	Parameter in Equation (9)	0.2	unitless

## Data Availability

The MATLAB code permitting to replicate our simulations is available on the repository Zenodo (https://doi.org/10.5281/zenodo.7271839).

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
