# Peer review of "Tortuous Cardiac Intercalated Discs Modulate Ephaptic Coupling"

_cells, 2022, doi:10.3390/cells11213477_

Round 1

Reviewer 1 Report

I have read the manuscript by Ivanovic and Kucera with great interest. The study uses computation to model geometric influences on the electrical signaling between the intercalated discs of cardiomyocytes. The results are interesting and will certainly spur new thoughts and directions for the growing group of scientists interested in ephaptic coupling. That being said, I do have some concerns and comments.

In general, isn’t the major impact of the geometries mediated via an effect on lateral resistance to the bulk interstitium (as stated in abstract and summary in introduction)? If so, it might be easier for the reader to keep this idea as a reference point during the presentation and discussion. Along these lines, a figure like figure 7A the radial folds may even be presented on a ‘negative’ axis in the same graph as the concentric folds, since they reflect lower Rlateral than the flat ID.

Just food for thought, since I must also confess that some of the text is a bit complicated to follow and might benefit from simplification in some cases and more explanation in others.

I also have some concern regarding the physiological relevance of the geometries applied, since none of them look like the structure of a plicate region (which I guess is the relevant structure when disregarding gap junctions). This need some discussion of the translational potential and be added as a major limitation in this regard. (if the geometry effect is mainly mediated by Rlateral, then it might not be so difficult to estimate the Rlateral of a more relevant geometry compare to flat?). The same arguments can be made regarding the concept of gathering all sodium channels in the middle of the disc.

The color-coding makes the figures difficult to follow, e.g. the varying nuances of purple/burgundy for larger N/Amp are hard to discern.

Below I have comment for specific elements of the manuscript in the order of appearance.

P1 abstract. I think a short summary of major limitations is in place (relevance of geometries, relevance of model with all sodium channels in center, clamped ion concentrations etc). Also, the term ‘in a complex manner’ is not very informative and could be elaborated.

P3

I think the present study only investigate some of the electric field aspects (Ve) of ephaptic coupling, while ephaptic coupling also includes changes in current due to altered ion concentrations (affecting currents both via altered channel conductance and Nernst potential).

End of first paragraph: it is not clear from text that references 18-21 and 11+22 are showing that essential conditions (defined in 6,7,9) are in place. Also, as I recall these conditions were already outline by Sperelakis, which should be acknowledged.

Apart from perinexi, clusters of sodium channels have also been shown around N-cadherin. Although the paper is cited, this issue should be considered as an alternative ‘ephaptic’ node.

Although we are now getting nice 3D structures of IDs, the tortuous structure of the plicate regions has been appreciated from EM for a long time and could be appreciated by reference.

Methods: some aspects are discussed in relation to results and limitations, see later. It is a bit difficult to decipher the overall makeup of the model. Is the intracellular node connected directly to all intracellular sites? I guess so since no intracellular resistivity is defined. Where then are the non-ID sodium channels placed, at the rim? Maybe an addition to figure 1 could clarify this.

P5

It has some logic that you need to scale the current to the relative area of the disc, but on the other hand the current density must have an effect of the cross talk across the cleft. This should be discussed and maybe the relative contribution be estimated at least for its direction.

P8

Please add some rationale for the choice of geometry (concentric and radial). Why did you not choose something closer to the plicate structure?

The abstract figure implies that the minimum Ve was in the middle of the ID. Was this the case for all time points in all tested scenarios and later figures (is the minVe curve essentially the data from the central node)? Assuming that Ina is the sum of sodium current from all nodes, then how much of the spread in time is explained by time course at the individual nodes and how much by delay in activation between nodes? This is particularly interesting for the -50 mV pulse where the time course varies greatly.

I guess that the increase in sodium current of cell two is eventually due to selfactivation (arrow III), but the text devotes little attention to the conditions that allow the first cell to trigger the sodium channels of cell two in the first place (transactivation). The latter may be of utmost importance from an ephaptic point of view.

There are two ephaptic processes, selfactivation and selfattenuation, which are likely to promote and reduce ephaptic coupling, respectively. Maybe the term ephaptic coupling needs to be defined explicitly. I interpret as the induction of transactivation or propagation of Aps, but please define. Increasing selfattenuation will, I guess, reduce the ephaptic coupling; so is there a combination where selfattenuation effectively prevents transactivation, e.g. at very narrow cleft size? The latter is in part addressed in figure 3b, but we only see the effect on Ve but not on peak Ina in cell 2.

P9

What is the physiological correlate if any of radial folds? Is it really ‘just’ a way of increasing and decreasing lateral resistance?

P14

Collecting all Ina in a central cluster is extreme and I wonder why you did not try to use something more realistic with many nodes scattered across the disc (eg. at one-two-three nodes spacing).

Also the change in protocol from voltage to current clamp is not clear.

The description of Na-transfer is well received, but since you do not model cleft sodium, my guess will be that you underestimate the current when cleft sodium goes down and subsarcolemmal goes down and the Nernst potential for sodium becomes more negative. This probably need some discussion and mention in limitations.

It seems that all conditions provoked an Ina response of a large magnitude in the neighboring cell except when clefts were wide and Rlateral low. This is an important effect since it indicates that transactivation in many cases is unavoidable, some explicit discussion is welcomed.

I guess the transmission of APs fails in most cases because the Rlateral is too high to permit threshold of activation of sarcolemmal channels? This could be stated more explicitly and set the stage for a discussion of the balance of shielding the nodes sufficiently to allow ephaptic mechanisms but not so too much so that the current can still affect the lateral membranes. If this is correctly understood then it could be stated much clearer.

P15

For radial folds the above is supported since the transactivation becomes even more sensitive to wide cleft size when R lateral drops.

P19

As noted above, I think more focus should be put on the conditions that allow transactivation of sodium channels, rather than on whether an AP is propagated or not. The reason for this being that the placement of all sodium channels in the middle of the ID is highly unnatural and the consequent shielding prevents activation of sarcolemmal sodium current. What is the bearing on real life propagation? Will the relationship be applicable to a situation with many more dispersed nodes and in particular with nodes closer to the lateral sarcolemma? This is a key point, because some might infer that ephaptic transmission of APs is a very fragile process that will fail in most instances. This needs to be discussed as well as stated in the limitations.

Along these lines may it also be an artifact that you do not see transfer extreme concentric folds 4-1Rid simply because you miss the point due to lack of resolution?

‘Note that for uniformly distributed Na channels and the same current clamp protocol….’ This is a very important point and quite counterintuitive. Any explanation for why this fails? Was this the true reason that you changed from voltage clamp? In that case, I think it should be moved up as a natural intro to why you changed protocol.

P21

Some of the discussion regarding clustering may have little bearing on a real ID since channels are known not to reside in a small cluster in the center. I would prefer a discussion of what elements we can actually use to predict the behavior of IDs.

The discussion of the relevance of the ID structure to arrhythmia could include the direction of effects in different situations. Most of the work cited implies that an increase in cleft width reduces conduction velocity, but the authors might also discuss how their work may explain the odd finding that in some pathological conditions, cleft widening may increase CV (PMIDs 30707595 & 32678707). The discussion has often been on the battle between self-activation and –attenuation, but the present really adds new information about a third party namely the lateral resistance to bulk.

As noted above, I also find the unnatural geometries to be a major limitation. Are the effects essentially boiled down to the Rlateral? If so can the Rlateral effect of a more plicate like geometry be calculated?

Also it stated many places that you test the effect of clusters, but in reality you only operate with one in center, which is highly unnatural. I fear that this is the reason why you predict that only a narrow range on conditions can produce transmission of APs. This is misleading in my opinion.

Using a linear conductance rather than one resembling Kir channels might also skew the results since depolarization will reduce the K current and thereby shunting along the ID. This would add to the problems outlined above.

Reviewer 2 Report

This is an interesting manuscript that seeks to determine how altering intercalated disc geometry and resistances impact ephaptic communication. There are 3 key findings. First, increasing radial resistance with concentric folds increases ephaptic interactions that are largely a function of path length (number and amplitude of folds similarly affected in concentric folds) while radial interactions decreased ephaptic responses.  When sodium channels were clustered in the center of the intercalated discs instead of being uniformly distributed, ephaptic interactions were complex. The manuscript is important and rich in both the data that is discussed and data that is not discussed. I’d like to congratulate the authors once again on an extremely thought provoking study.

I did struggle with a core concept, and that is the “level” of ephaptic coupling. I will focus most of my comments on figures 2 and 4 as the exemplars. I do not know if answering these questions will add clarity to whether ephaptic coupling has an amplitude, but it may enrich the understanding of ephaptic communication.

The authors communicate that the ephaptic levels are based on the changes in extracellular potential and peak sodium channel current. Could the authors discuss the concept of ephaptic levels when self-activation and attenuation oppose each other and self-activation appears to fail under conditions that should support enhanced ephaptic self-activation? For example, in figure 2, the suprathreshold stimulus (-25mV) produces a larger peak sodium current in cell 1 than a near-threshold (-50mV) stimulus. Yet, the spatial minimum Ve is larger in the -50mV case because sodium channels in cell 2 activate and further decrease Ve.

Can the authors discuss why the -25mV stimulus does not self-activate sodium channels in cell 2 despite the larger and faster activating currents in cell 1? This is also observed in the radial folds simulation in figure 4, and this is a surprising result upon first inspection.

Is it possible to state which case is more ephaptic in Figure 2 – left or right panels? The stimulus threshold appears to play a significant role in the ephaptic response. I recognize that this begins to delve into the realms of “safety factor,” and I think it is ok to leave that discussion out at this stage.

It appears that there are lower bounds to self-activation. In this and the author’s 2018 manuscript, the peak currents in the proximal cell appear fairly comparable in the -25 and -50mV stimulus cases. The difference between the change in min Ve for Figure 2, left and right panel, are the durations of the sodium transient and potentially the time needed to recruit enough channels to self-activate the channels in cell 2.  But in the radial fold case (Figure 4), the longer activating -50mV case does not lead to a noticeable response in cell 2. Can the authors comment on the drivers of self-activation – peak current amplitude, time-course of activation, or is Figure 2 (-50mV) showing a radially propagated membrane response?

The data is very intriguing because it suggests that conduction failure at some narrow or tortuous cleft widths may not due to self-attenuation following self-activation by a rapid collapse of the driving force, but rather a failure of self-activation in cell 2.

Minor comments.

Can the authors plot the peak sodium current in an isolated cell without ephaptic coupling?

In a few places, the authors make the comment that outward INa provides the cleft with Na+ ions, but then note in the discussion that this model assumes constant extracellular ion concentrations. In this all electrical model, will it be more accurate to say that outward INa provides the cleft with more positive charge?

The color schemes, dotted, and dotted dashed lines are difficult to distinguish. For example, N=8 is very similar in color to N=4, and the dotted lines in figure 5A –INa were also difficult to distinguish.

Please describe min Ve in the methods or early in the results for those unfamiliar with the authors’ previous works.

Reviewer 3 Report

The authors ingeniously designed a computer model to simulate the relevance of tortuous intercalated discs on ephaptic coupling. Could the corresponding experimental evidence be provided? The authors previous modeling studies focused on the distribution of Na+ channels and gap junctions in intercalated discs on ephaptic coupling. Is it possible to establish a complete model for exploring these multifactors regulation on ephaptic coupling? 

Round 2

Reviewer 1 Report

The authors' response and modifications to the manuscript addresses my concerns. I congratulate the authors with an interesting paper and look forward to future studies from the group.